# A Highly Flexible Piezoelectric Ultrasonic Sensor for Wearable Bone Density Testing

**DOI:** 10.3390/mi14091798

**Published:** 2023-09-20

**Authors:** Zhiqiang Song, Bozhi Wang, Zhuo Zhang, Yirong Yu, Dabin Lin

**Affiliations:** 1Department of Automation and Robotics Engineering, School of Automation, Wuxi University, Wuxi 214105, China; zqsong@cwxu.edu.cn; 2School of Optoelectronic Engineering, Xi’an Technological University, Xi’an 710032, China; 19591526681@163.com (B.W.); zhuozhang2021@163.com (Z.Z.);

**Keywords:** piezoelectric materials, bone density testing, wearable sensor, flexible ultrasonic device

## Abstract

Driven by the loss of bone calcium, the elderly are prone to osteoporosis, and regular routine checks on bone status are necessary, which mainly rely on bone testing equipment. Therefore, wearable real-time healthcare devices have become a research hotspot. Herein, we designed a high-performance flexible ultrasonic bone testing system using axial transmission technology based on quantitative ultrasound theory. First, a new rare-earth-element-doped PMN-PZT piezoelectric ceramic was synthesized using a solid-state reaction, and characterized by X-ray diffraction and SEM. Both a high piezoelectric coefficient *d*_33_ = 525 pC/N and electromechanical coupling factors of *k*_33_ = 0.77, *k*_t_ = 0.58 and *k*_p_ = 0.63 were achieved in 1%La/Sm-doped 0.17 PMN-0.47 PZ-0.36 PT ceramics. Combining a flexible PDMS substrate with an ultrasonic array, a flexible hardware circuit was designed which includes a pulse excitation module, ultrasound array module, amplification module, filter module, digital-to-analog conversion module and wireless transmission module, showing high power transfer efficiency and power intensity with values of 35% and 55.4 mW/cm^2^, respectively. Finally, the humerus, femur and fibula were examined by the flexible device attached to the skin, and the bone condition was displayed in real time on the mobile client, which indicates the potential clinical application of this device in the field of wearable healthcare.

## 1. Introduction

The internet of things (IOT) is a rapidly growing industry, and its potential in improving human quality of life has led to its adoption in multiple industries, with healthcare being one of the most promising [1,2,3,4,5]. The internet of wearable things (IoWT) is technology with the potential to revolutionize the healthcare industry through automated remote healthcare [6,7,8,9,10]. Wireless sensors connected to wearable devices continuously monitor human activities and health factors, and collect data to enable clinical doctors to remotely access patients [11]. With an increase in age, the loss of bone nutrients due to various reasons leads to the decline of bone density and bone quality, and the damage of bone microstructure to a certain extent, resulting in increased bone fragility and the increase of fracture risk [12]. Therefore, finding a method to diagnose osteoporosis rapidly and effectively at an early stage has become an important research direction in the medical field [13,14,15].

Currently, there are several bone detection systems that can determine bone quality in certain parts of the human body, including dual-photon absorptiometry (DPA) [16], single-photon absorptiometry (SPA) [17], Dual-Energy X-ray Absorptiometry (DEXA) [18] and quantitative CT (QCT) [19], which is clinically limited due to its high radiation dose, despite its high accuracy. Considering the cost of the instrument and the radiation intensity of the body, quantitative ultrasound (QUS) is one of the best methods to diagnose osteoporosis [20], which is overcome the limitations of the methods based on X-rays. This method can detect the bone condition by transmitting ultrasonic waves to the measured site and receiving echoes.

Wearable sensors are emerging as a research hotspot due to their combination of biocompatibility, high elasticity and stretchability [21,22,23,24]. Flexible sensors convert physiological signals into electrical signals in the form of signal transduction and have great potential in human health detection, biomedicine and flexible electronic skin [25,26,27,28]. Realizing the multi-functionality, comfort and accuracy of flexible electronic devices could contribute to the development of wearable medical devices, and also promote the generation of medical devices that integrate human disease prediction, analysis and diagnosis. Hong et al. combined piezoelectric ceramics with PDMS flexible materials to prepare a kirigami-structured highly anisotropic piezoelectric network composite sensor for preventing joint disorders and detecting joint motion [21]. Jin et al. prepared ultra-thin flexible printed circuits using flexible printed circuit technology, welded ultrasonic sensors onto prefabricated circuits, and used flexible-based silicone (Ecoflex) as packaging. A flexible ultrasonic energy transmission device and a flexible Doppler ultrasonic device for monitoring blood flow velocity were investigated [29].

Piezoelectric materials are the core part of ultrasonic transducers [30,31]. In this work, piezoelectric materials with high electromechanical coupling factors and low dielectric loss were selected for the preparation of a high-precision flexible ultrasonic array. Based on quantitative ultrasound theory [20,32], a new type of rare-earth-element-doped PMN-PZT ceramic was used to design a high-sensitivity flexible ultrasonic (HSFU) sensor, and a high-performance flexible ultrasonic bone density measurement system based on axial transmission technology was designed to detect bone conditions in different parts of the human body. Firstly, the high piezoelectric properties of La/Sm-doped PMN-PZT ceramics were prepared, and a piezoelectric ultrasonic array was designed as the core sensor element. Secondly, the flexible system hardware was designed, including a flexible sensor, power supply circuit, pulse excitation circuit, gain control circuit, A/D conversion circuit, FPGA minimum system design and wireless transmission module. Finally, the visual man–machine interface was designed on the mobile terminal, displaying the bone condition of the humerus, femur and fibula, including the test values of SOS, T and Z.

## 2. Materials and Methods

### 2.1. Piezoelectric Materials

La/Sm-Pb(Mg_1/3_Nb_2/3_)O_3_-Pb(Zr,Ti)O_3_ (La/Sm-PMN-PZT) piezoelectric ceramics were prepared using a B site cation precursor method [33,34]. As shown in Figure 1a, the MgNb_2_O_6_ precursor materials were first fabricated at 1200 °C for 4 h using Nb_2_O_5_ and MgCO_3_ powders. The Pb_3_O_4_, MgNb_2_O_6_, ZrO_2_ and TiO_2_ powders were wet-mixed using zirconium ball milling with alcohol as a solvent for 6 h. Second, the mixed powders were calcined at 850 °C for 2 h and vibratory-milled in alcohol with a binder for 6 h. After drying at 80 °C for 10 h, the powders were pressed into pellets 5 mm in thickness and 20 mm in diameter under the uniaxial pressure of 500 MPa, respectively. At 550 °C, the binder was burned out over 2 h, and the samples were sintered in sealed corundum crucibles at 1200~1250 °C for 2 h.

The crystal structure of samples was determined by XRD data collected via X-ray diffraction (Bruker D8, Blue Science, LLC, The Colony, TX, USA). The morphology of samples was measured by a field emission scanning electron microscope (Zeiss Gemini SEM 500, Zeiss, Jena, Germany). For further electric measurement, silver paste was fired on both sides of the samples at 600 °C for 10 min to form the electrodes. The samples were poled in silicone oil at 100 °C for 30 min using a DC electric field with a strength of 20 kV/cm. The piezoelectric coefficients were determined by a quasi-static d_33_-m. The temperature-dependent dielectric properties were determined using an LCR meter (HP 4284 A, Hewlett-Packard GmbH, Koto-ku, Tokyo) connected to a computer-controlled cooling–heating stage. The electromechanical coupling factors *k*_33_, *k_t_* and *k_p_* values were determined by the resonance method employing an impedance analyzer (HP 4194 A, Hewlett-Packard GmbH) according to IEEE standards on piezoelectricity, according to the following formula:(1)$$k_{p}^{2}=\frac{1}{p}\frac{f_{a}^{2}-f_{r}^{2}}{f_{a}^{2}}$$
(2)$$k_{t}^{2}=\frac{\pi }{2}\frac{f_{r}}{f_{a}}cot(\frac{\pi }{2}\frac{f_{r}}{f_{a}})$$
(3)$$k_{33}^{2}=k_{p}^{2}+k_{t}^{2}-k_{p}^{2}k_{t}^{2}$$
where *f_r_* is the impedance spectra of resonant frequency, and *f_a_* is the anti-resonant frequency of the sample.

### 2.2. Fabrication of Piezoelectric Sensor

The piezoelectric arrays were developed in this work using La/Sm-doped PMN-PZT ceramic films as a piezoelectric layer and PDMS as a flexible countersink [35]. The fabrication process is presented in Figure 1b. First, SiO_2_ with a thickness of 1µm was embedded on silicon and the top silicon layer was 15 µm, as shown in Figure 1b—I. Chromium (Cr) and gold (Au) electrodes with thicknesses of 200 nm and 500 nm, respectively, were magnetron-sputtered on both sides of the doped PMN-PZT ceramics with thickness of 200 μm, bonded to a silicon-based wafer, as shown in Figure 1b—II, and the structure was placed on a wafer bonding machine, heated to a bonding temperature of 160 °C, and a pressure of 3.5 kgf/cm was applied. The thickness of the doped PMN-PZT ceramics was reduced to 50 µm using plasma etching, and the top electrode Pt/Ti (300 nm) was sputtered and patterned by a stripping process (Figure 1b—III). The ceramics were etched in etchant until the bottom electrode Cr was exposed (Figure 1b—IV). After ceramic etching, a 300 nm gold film bonding layer was deposited and patterned using a stripping process to provide an electrical connection to the bottom electrode (Figure 1b—V). Then, a 100 nm thick Al layer was deposited as an etch mask for the deep reactive ion etching process using wet etching for patterning on the backside of the Si substrate. The etching process terminated at the SiO_2_ layer (Figure 1b—VI). To fabricate the flexible device, polydimethylsiloxane (PDMS) was used in this paper and highly bonded to the wafer (Figure 1b—VII).

### 2.3. Bone Density Testing Method

The system uses an ultrasound array with single transmitting and dual receiving as the front-end signal collector. There are three ultrasonic array sensors, A, B and C, located on the same horizontal line, where A is used as transmit ultrasonic waves and B and C are used to receive ultrasonic waves. The schematic diagram of the ultrasonic array detection is shown in Figure 1c. When the ultrasonic transducer transmits ultrasonic waves, there will be a wave beam incident in the direction of the critical angle into the bone tissue, generating a side wave transmitted along the inner surface of the bone and refracted out of the bone at an equal angle, and the sensor at the receiving end is responsible for receiving the signal.

The system records the ultrasonic transmission time between each transmitter and receiver. A emits ultrasound and B receives the signal, recording the transmission time *T_AB_* of the acoustic wave between A and B. A emits ultrasound and C receives the signal, recording the transmission time *T_AC_* of the acoustic wave between A and C.

From Figure 1c, it can be seen that:*T_AB_ = t*_1_ + *t*_2_ + *t*_3_(4)
*T_AC_ = t*_1_ + *t*_2_ + *t*_4_ + *t*_5_(5)
since the planes in which A, B and C are parallel to the surface of the bone. When the ultrasound waves emitted pass through the soft tissue at equal times and distances $t_{3}$ = $t_{5}$, *S_BB_*_1_ = *S_CC_*_1_, then:(6)$$\mathrm{SOS}=\;\frac{d}{t_{4}}\;=\frac{d}{T_{AC}-T_{AB}}$$

From Equation (6), when A, B and C are in planes parallel to the surface of the bone, it is only necessary to know the transmission time of the ultrasound waves from A to B and C, respectively, to determine the ultrasound sound velocity *SOS* in the bone.

The T-Score and Z-Score are used as diagnostic criteria for osteoporosis. The T-Score is the result of comparing the bone mass of the test subject with that of a young person of the same sex. The specific formula for its calculation is as follows:(7)$$T-Score=\frac{SOS-\bar{SOS}}{SD}$$
(8)$$SD=\sqrt{\sum_{i=1}^{n}{\left(SOS_{i}-\bar{SOS}\right)}^{2}}$$
where:

*SOS*—Ultrasound velocities at specific skeletal sites in subjects.

$\bar{SOS}$—Reference standard values.

*SD*—Standard deviation.

The values of $\bar{SOS}$ and *SD* are related to the bone densitometer used for the measurement, the measurement site, the ethnicity, the gender and the sample population selected, and the selection of different databases will result in different $\bar{SOS}$ and *SD. Z-Score* indicates the result of comparing the bone mass of the subject with that of a healthy person of the same age and sex. The *Z-Score* and *T-Score* are calculated in the same way; only the reference standard chosen is different.

## 3. Results and Discussion

### 3.1. Piezoelectric Properties of La/Sm-PMN-PZT

Piezoelectric materials with novel piezoelectric response are the core of sensor arrays, providing high energy conversion efficiency [36]. The testing system is made highly flexible by placing three sensor arrays at various angles. In addition, the size expansion allows the system to obtain the speed of ultrasound propagation through the bone material in order to comprehensively measure the bone condition.

The XRD patterns of PMN-PZ-PT ceramics with 1.25 mol% and 1 mol% La/Sm dopant are shown in Figure 2a,b, respectively. No spurious peaks appear in the range of 20° to 80°, while double-doping the rare earth elements La and Sm does not change their phase structure. The prepared ceramics are all pure perovskite phase without any impurity phases such as pyrochlore phase [33]. The SEM images of the untreated natural surfaces of 1.25 mol% and 1 mol% La/Sm-doped PMN-PZT ceramics are listed in Figure 2c,d. There are almost no pores in all samples, which present uniform grain and high density.

The piezoelectric properties of PMN-PZ-PT ceramics prepared by introducing Zr into PMN-PT are significantly enhanced. Meanwhile, the rare earth elements La and Sm are added to the co-doping modification in order to obtain high-quality ceramics with good electrical properties. The piezoelectric and dielectric properties of the different components of the PMN-PZ-PT ceramics with La/Sm co-doping are listed in Table 1. The highest value of piezoelectric constants is 525 pC/N, found in 1%La/Sm-doped 0.17 PMN–0.47 PZ–0.36 PT ceramics, as shown in Figure 2e. The good piezoelectric properties of the samples are closely related to the high density and uniform micron grain size [37]. The doped PMN-PZT presented higher values of piezoelectric constants and electromechanical coupling factors than that of pure PMN-PT and PZT ceramics, as shown in Figure 2f,g. In addition, rare element doping improves the ductility of the sample and shows better tensile properties than PZT, as shown in Figure 2h.

### 3.2. HSFU Design

The manufacturing process of the ultrasonic array is shown in Figure 1b. The arrays were placed in the mold and the PDMS liquid was dropped. Then, it was placed in a vacuum drying oven and cured at 50 °C for 10 h to complete the preparation of the HSFU arrays. Here, three ultrasonic arrays and interconnects were packaged in PDMS, which makes the ultrasonic arrays soft and adaptable to most body parts, such as the skin surface of the arm, as shown in Figure 3a. Based on the transverse transmission characteristics of ultrasound through the superficial bone layer, the angles between the three groups of ultrasonic arrays and the plane are different, which are in the orders of 17°, 20° and 23°, respectively. The cross-section view is shown in Figure 3b. As shown in Figure 3c, after receiving the pulse of the excitation circuit, the ultrasonic wave is emitted by the ultrasonic emission array, which is refracted into two ultrasonic receiving rays after propagation in the bone. The receiving array converts the acoustic signal into an electrical signal using the piezoelectric effect, and the generated electrical signal is continuously transmitted to the intelligent device through the field programmable gate array (FPGA) for data analysis and storage. Flexible circuits can be designed into complex three-dimensional structures, bent into a variety of shapes and can be used for highly repetitive applications. Under different deformation states, HSFU can still perform corresponding electrical functions, and can completely transmit pulse signals to complete digital-to-analog signal conversion as shown in Figure 3d,e.

### 3.3. Ultrasonic Energy Transfer

For piezoelectric sensors, high-frequency electrical signals are generated internally after high-frequency vibration is generated. On the other hand, when a high-frequency electrical signal is added to the piezoelectric ceramic, it will produce a high-frequency mechanical vibration signal, so it is necessary to study the energy transfer efficiency of the sensor. As shown in Table 2, the response peak voltage is 2.26 V under the 8 V excitation peak voltage in the water transmission medium when the transmission distance is 5 mm.

The electrical connection diagram of the sensor is shown in Figure 4a. The internal charge accumulation of piezoelectric material after excitation and after receiving ultrasound is presented in Figure 4b. After the sensor was cured, the resonance frequency was slightly shifted from 2 MHz to 2.11 MHz, as shown in Figure 4c. For calculating the excitation power P_in_, the waveforms of excitation voltage and current are shown in Figure 4d. For calculating the received power P_out_, the waveform of response voltage is presented in Figure 4e. Under various excitations, the response voltage of the sensor dependent on frequency is shown in Figure 4f. Under different distances, the changes in response voltage dependent on frequency and excitation are presented in Figure 4g,h, respectively. With water as the transmission medium, the values of input power, received power and power intensity are 208 mW, 72 mW and 55.4 W/cm^2^, respectively. Furthermore, the power transmission efficiency of the sensor can reach up to 35%.

### 3.4. Hardware Design and Wireless Transmission

The system integrates six modules: a pulse excitation module, ultrasonic array module, amplifier module, filter module, digital-to-analog conversion module and wireless transmission module, as shown in Figure 5a. First, the main control chip (FPGA) makes the pulse excitation module generate high-frequency pulse signals, so that the transmitting port sends ultrasonic waves, which are transmitted through the skeleton and received by the receiving port. Second, the receiving port converts sound energy into an electrical signal, the filter circuit filters the signal and the amplifier circuit amplifies the signal. The analog signal is converted into a digital signal by the A/D conversion circuit, and enters FPGA for processing and analysis. Finally, it is transmitted through the wireless transmission module and displayed on the mobile terminal. In addition to the ultrasonic array module, the remaining five modules are integrated on flexible PCB, as shown in Figure 5b.

In order to test wireless communication, the signal generator provided a sine wave with a frequency of 1.25 MHz, and after 2 s, the input signal of the second channel was calculated to set the speed of 0.75 mm/s, as shown in Figure 5c. With a step length of 0.7 s, the time interval gradually changed from 2 to 5.5 s, as shown in Figure 5d. The set speed and measured speed were recorded, and the average relative error of the result was 0.3%. Then, the signal generator provides a FPGA square wave and triangle wave successively, as shown in Figure 5e. The FPGA encodes the information and calculation results of the two waveforms into 10-bit binary data through UART. The APP receives information from the WIFI module through the UDP protocol and displays each data point to the corresponding position of the APP through decoding, as shown in Figure 5f, and obtains a real-time view function.

### 3.5. Real-Time Display of Bone Testing

To verify the performance of entire HSFU sensors, the flexible system was fixed on the thigh, as shown in Figure 6a. The measured sensor signal and calculated values of SOS, T-Score and Z-Score of the femur were in the orders of 3404.46 m/s, −0.1 and −0.2, as shown in Figure 6a, respectively. A T-Score greater than −1 is normal, and there is no bone loss or osteoporosis. When the T-Score is between −1 and −2.5, it indicates bone loss, but it does not reach the degree of osteoporosis [38,39]. A T-Score of less than −2.5 is associated with osteoporosis. The current subject is a young male aged 25 years, so the T-Score should be greater than −1, consistent with the test values.

Due to the different number of blood vessels and soft tissues in different parts of the human body, different reflections and refractions of the ultrasonic wave in the body are encountered when propagating, and the signal ultimately entering the receiver is also changed [40,41,42]. When the system is fixed on the arm and calf, the waveform obtained by the receiving port is different, as shown in Figure 6b,c. For humerus testing, the values of SOS, T-Score and Z-Score were in the orders of 3814.11 m/s, 0.2 and 0.3, as shown in Figure 6b. For fibula testing, the values of SOS, T-Score and Z-Score were in the orders of 3298 m/s, −0.8 and −0.6, as shown in Figure 6b. In general, these results show that there are slight differences in bone density in different parts of the same human body, which also verifies the accuracy of the wearable system.

## 4. Conclusions

In this work, we demonstrate a flexible ultrasonic sensor and design a wearable ultrasonic bone density detection system based on the novel piezoelectric characteristics of PMN-PZT ceramics, which has the advantages of multi-site measurement and instantaneous detection capability. After rare-earth-element doping, the PMN-PZT ceramics showed excellent dielectric and piezoelectric properties, and the Curie temperature and electromechanical coupling factor *k*_33_ reached 255 °C and 0.77, respectively. Compared with conventional ultrasonic devices, the developed flexible ultrasonic sensors have a high degree of flexibility, making them well adapted to multiple parts of the human body. In the hardware circuit, the synchronous scanning circuit provided a single frequency of 1.15 MHz, and a 5 V signal for the pulse excitation circuit, which controlled the ultrasonic sensor to transmit the detection signal with a frequency of 1.15 MHz and an 80 V pulse excitation signal; the gain control circuit amplified the echo signal to make it 1 V~2 V, which was convenient for the sampling of A/D conversion circuit; and the FPGA adopted a minimum system design and realized wireless data transmission through the wireless transmitting module. Finally, the mobile APP presented the measurement results of the humerus, femur and fibula in real time, including SOS, Z and T values.

## Figures and Tables

**Figure 1 micromachines-14-01798-f001:**
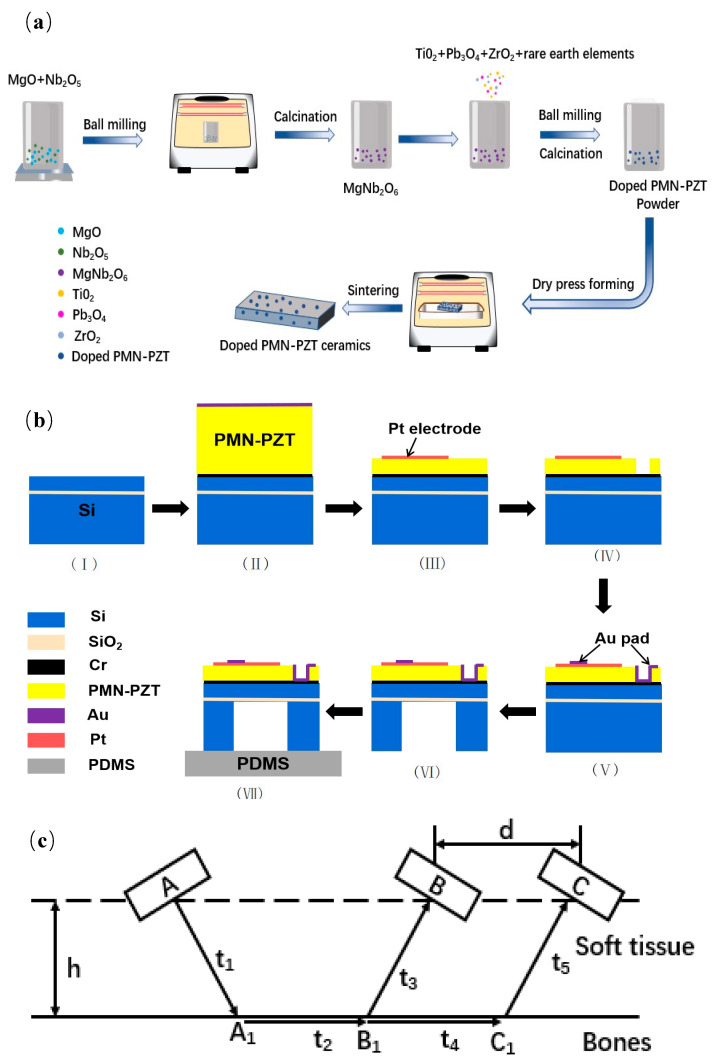
Schematic diagram of piezoelectric ceramic preparation, ultrasonic sensor fabrication and bone testing method. (**a**) Synthesis process of rare-earth-element-doped PMN-PZT piezoelectric ceramics; (**b**) process of ultrasonic sensor fabrication: (I) Si substrate with 1 µm SiO_2_, (II) bonding of doped PMN-PZT ceramics on Si, (III) top electrode patterning (Pt/Ti) after PMN-PZT reduction, (IV) PMN-PZT etching for electrical connection of bottom electrode, (V) Au deposition and patterning for electrical connection, (VI) releasing from the back-side using DRIE process, and (VII) bonding of the Si on flexible PDMS layer; (**c**) measuring method of bone testing.

**Figure 2 micromachines-14-01798-f002:**
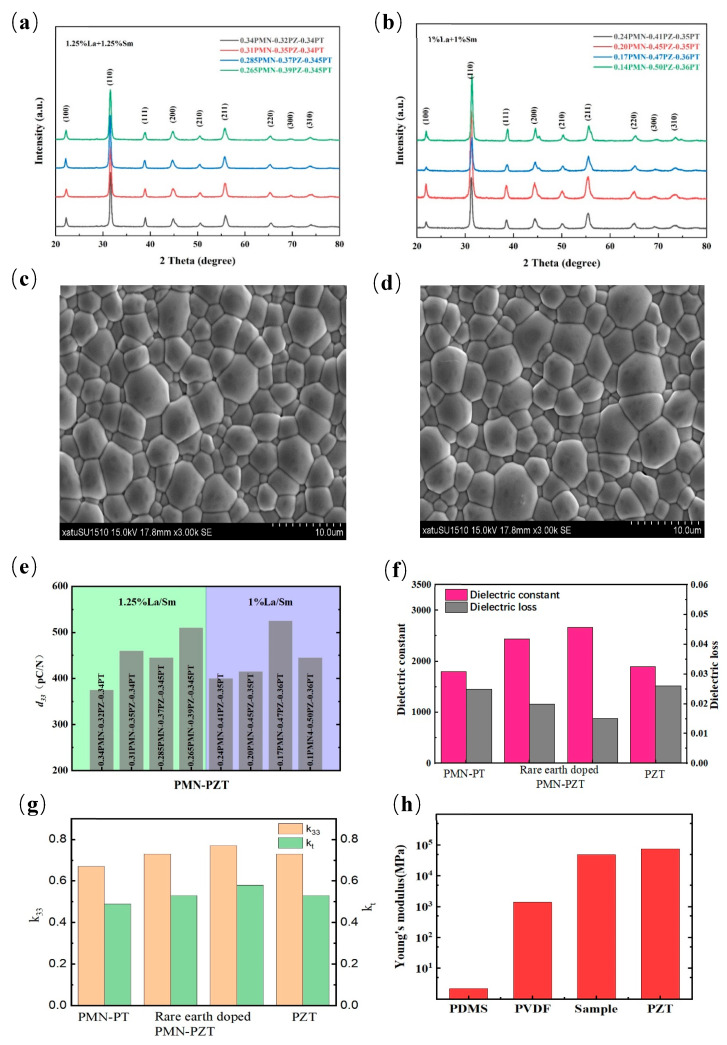
Structure and electric properties of La/Sm-doped PMN-PZT piezoelectric ceramics: (**a**) XRD pattern of 1.25% La/Sm-doped PMN-PZT ceramics, (**b**) XRD pattern of 1% La/Sm-doped PMN-PZT ceramics, (**c**) SEM diagram of 1.25% La/Sm-doped PMN-PZT ceramics, (**d**) SEM diagram of 1% La/Sm-doped PMN-PZT ceramics, (**e**) the *d*_33_ values of 1% La/Sm-doped PMN-PZT ceramics with various compositions, (**f**) the dielectric constant and dielectric loss of 1% La/Sm-doped PMN-PZT compared with PMN-PT and PZT, (**g**) the *k*_33_ and *k*_t_ values of samples compared with PMN-PT and PZT, and (**h**) Young’s modulus of samples compared with PDMS, PVDF and PZT.

**Figure 3 micromachines-14-01798-f003:**
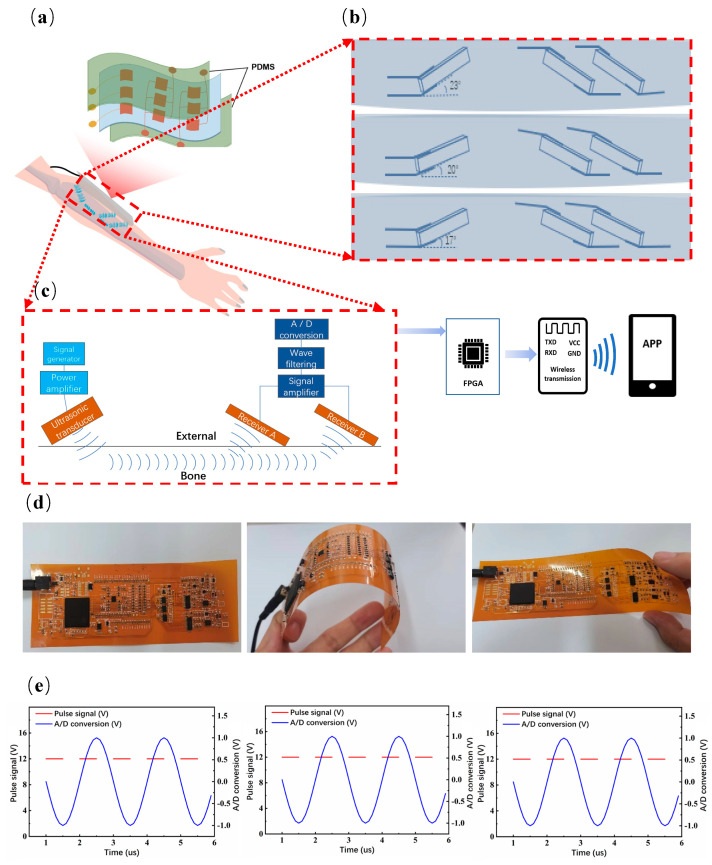
Schematic and flexible circuit HSFU sensor: (**a**) Schematic of flexible bone testing device; (**b**) cross-sectional views of ultrasonic arrays with tilt angles of 17°, 20° and 23°, respectively; (**c**) system architecture of HSFU sensor; (**d**,**e**) the pulse signal generated by the pulse excitation circuit and operation of the A/D conversion circuit in the flattened state, bent state and twisted state of the flexible circuit.

**Figure 4 micromachines-14-01798-f004:**
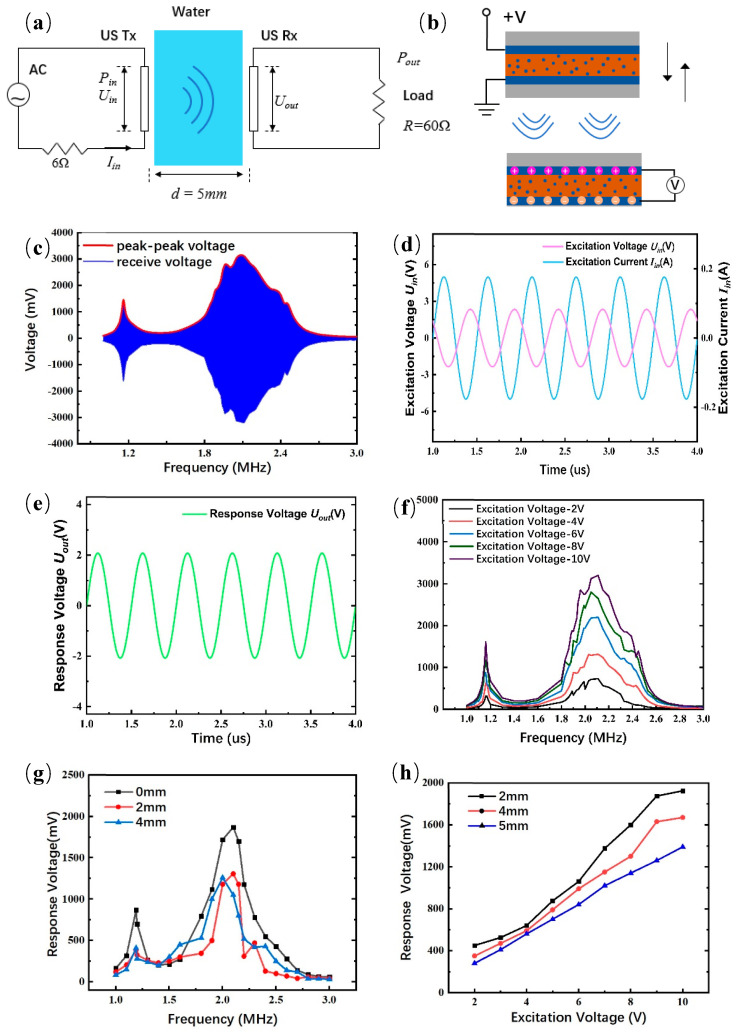
Testing of ultrasonic sensor: (**a**) Electrical connection diagram of ultrasound sensor, (**b**) schematic of ultrasonic sensor during transmission and reception, (**c**) sensor response voltage testing, (**d**) measured electrical waveforms of excitation voltage U_in_ and current I_in_ for calculating consumed power P_in_, (**e**) measured electrical waveform of response voltage of U_out_ for calculating received power P_out_, (**f**) response voltage of sensor with changing frequencies under different excitation, (**g**) response voltage of sensor with changing frequencies at various distances, and (**h**) response voltage of sensor with changing excitation at various distances.

**Figure 5 micromachines-14-01798-f005:**
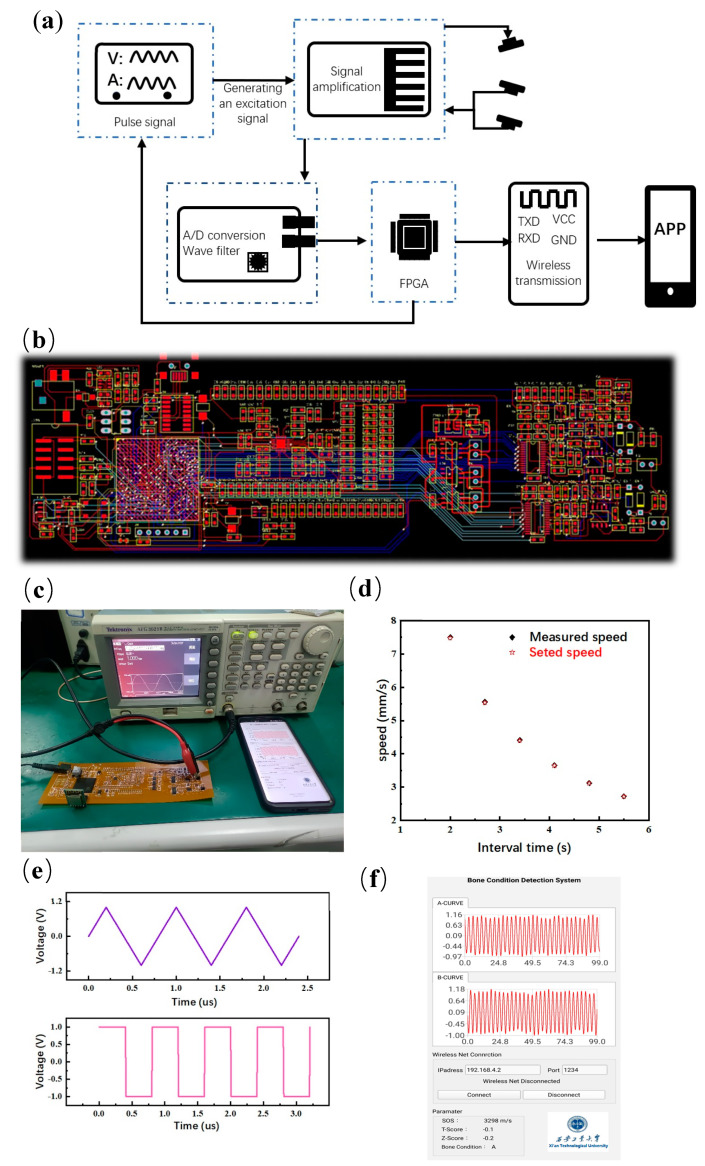
Hardware design and wireless transmission: (**a**) Structure diagram of bone testing system, (**b**) PCB circuit design and product drawing, (**c**) hardware testing process demonstration, (**d**) testing results of sound velocity, (**e**) testing results of wireless transmission, and (**f**) APP interface display window.

**Figure 6 micromachines-14-01798-f006:**
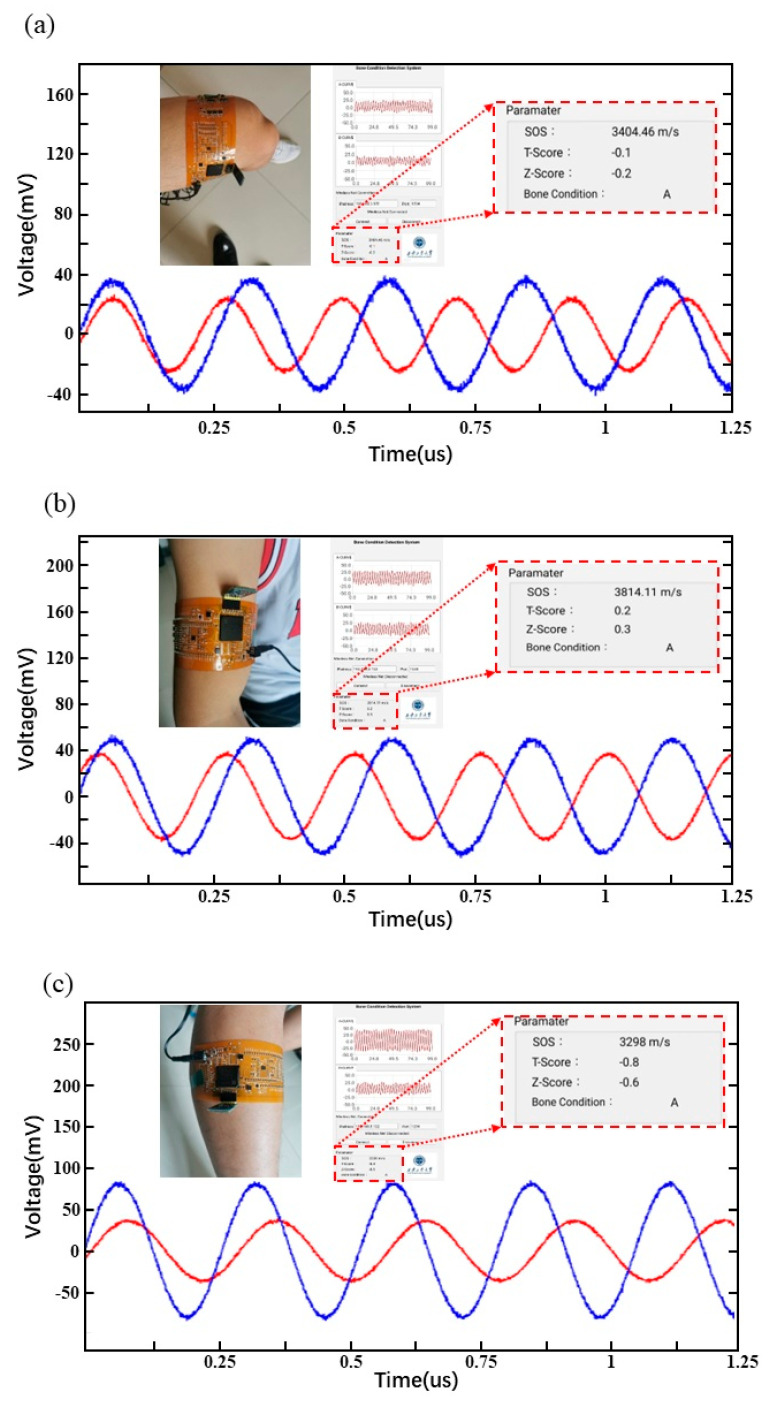
Human body testing and real-time display: (**a**) femur, (**b**) humerus, (**c**) fibula.

**Table 1 micromachines-14-01798-t001:** Piezoelectric and dielectric properties of La/Sm-PMN-PZT ceramics.

La/Sm-PMN-PZ-PT	*T*_c_ (°C)	*d*_33_ (pC/N)	*ε* _r_	*k* _p_	*k* _t_	*k* _33_
0.0125/0.34/0.32/0.34	187	380	2262	0.54	0.48	0.67
0.0125/0.31/0.35/0.34	196	460	2347	0.58	0.51	0.71
0.0125/0.285/0.37/0.345	212	445	2353	0.59	0.52	0.72
0.0125/0.265/0.39/0.345	218	510	2442	0.59	0.53	0.73
0.01/0.24/0.41/0.35	221	400	2147	0.61	0.53	0.74
0.01/0.20/0.45/0.35	234	415	2053	0.61	0.54	0.74
0.01/0.17/0.47/0.36	255	525	2676	0.63	0.58	0.77
0.01/0.14/0.50/0.36	264	445	1947	0.59	0.57	0.74

**Table 2 micromachines-14-01798-t002:** Response voltage depends on transmission depth.

Transmit Deepness (mm)	1	2	4	5
Response Peak-Peak Voltage (mV)	1420	1180	1980	2280
Response Voltage Root Mean Square (mV)	626	531	893	979
Excitation Peak-Peak Voltage (mV)	2000	4000	6000	8000
Media Layer	Water	Water	Water	Water

## Data Availability

Not applicable.

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
