# Peer review of "A Highly Flexible Piezoelectric Ultrasonic Sensor for Wearable Bone Density Testing"

_micromachines, 2023, doi:10.3390/mi14091798_

Round 1
Reviewer 1 Report
This manuscript is excellently written, and informative, and the research presented demonstrates both high interest and quality. I strongly recommend publication.
My only suggestion is about the abstract. It does not contain the keywords of this research as it should. While you can find some details about the device, there is not much information about the mechanism that the device works based on. The abstract should encompass all crucial information about the research, which is currently lacking in this one.
Reviewer 2 Report
The manuscript is dedicated to study of a flexible ultrasonic piezoelectric ultrasonic sensor for bone density testing. The proposed PMN-PZT ceramic with rare earth elements doping exhibited remarkable dielectric and piezoelectric properties; the synchronous scanning circuit controlled the ultrasonic sensor to transmit the detection signal, successfully demonstrating the measurement results of humerus, femur and fibula in real time. The manuscript is generally well-written and sound but still with a number of typos and grammar issues that need carefully amendments. Also note that unclear, small labels in Fig. 2a, 2b, 2e, 3c, 3e should be revised properly. Moreover, in line 220, the author should explain the reason that the three groups of ultrasonic arrays need to be placed on the order of 17 degree, 20 degree, and 23 degree, respectively.
The manuscript is generally well-written and sound but still with a number of typos and grammar issues that need carefully amendments.
